# Decreased Thrombospondin-1 and Bone Morphogenetic Protein-4 Serum Levels as Potential Indices of Advanced Stage Lung Cancer

**DOI:** 10.3390/jcm10173859

**Published:** 2021-08-27

**Authors:** Monika Kosacka, Tomasz Dyła, Monika Chaszczewska-Markowska, Katarzyna Bogunia-Kubik, Anna Brzecka

**Affiliations:** 1Department of Pulmonology and Lung Oncology, Wroclaw Medical University, 53439 Wroclaw, Poland; tomasz.dyla@umed.wroc.pl (T.D.); anna.brzecka@umed.wroc.pl (A.B.); 2Laboratory of Clinical Immunogenetics and Pharmacogenetics, Ludwik Hirszfeld Institute of Immunology and Experimental Therapy, Polish Academy of Sciences, 53114 Wroclaw, Poland; monika.chaszczewska-markowska@hirszfeld.pl (M.C.-M.); katarzyna.bogunia-kubik@hirszfeld.pl (K.B.-K.)

**Keywords:** lung cancer, biomarkers, thrombospondin-1, bone morphogenetic protein-4

## Abstract

Introduction: Lung cancer belongs to the most common carcinoma worldwide and is the leading cause of cancer-related death. Bone morphogenetic protein-4 (BMP-4) is extracellular signaling molecule involved in many important processes, including cell proliferation and mobility, apoptosis and angiogenesis. Thrombospondin-1 (TSP-1) belongs to the extracellular matrix proteins. It participates in the cell-to-cell and cell-to-matrix interactions and thus plays important role in tumor microenvironment for cancer development and metastasis formation. Aim: To investigate serum levels of TSP-1 and BMP-4 together with BMP-4 polymorphism in lung cancer patients. Material and Methods: A total of 111 patients (76 men) with newly diagnosed lung cancer, including 102 patients with non-small cell lung cancer and 9 patients with small-cell lung cancer. Advanced stage of lung cancer was diagnosed in 99 (89%) of patients: stage IV—in 48, stage IIIB—in 33, stage IIIA—in 18 patients; there were six patients with stage II and six patients with stage I. The control group consisted of 61 healthy persons. In all the subjects, serum levels of BMP-4 and TSP-1 were measured by ELISA. With a Real-Time PCR system genotyping of BMP-4 was performed. Results: BMP-4 and TSP-1 serum levels were significantly lower in the patients with lung cancer than in the controls (TSP-1:10,109.2 ± 9581 ng/mL vs. 11,415.09 ± 9781 ng/mL, *p* < 0.05; BMP-4: 138.35 ± 62.59 pg/mL vs. 226.68 ± 135.86 pg/mL *p* < 0.001). In lung cancer patients TSP-1 levels were lower in advanced stages (9282.07 ± 4900.78 ng/mL in the stages III-IV vs. 16,933.60 ± 6299.02 ng/mL in the stages I-II, *p* < 0.05) and in the patients with than without lymph nodes involvement (10,000.13 ± 9021.41 ng/mL vs. 18,497.75 ± 12,548.25 ng/mL, *p* = 0.01). There was no correlation between TSP-1 and BMP-4 serum levels. BMP-4 gene polymorphism did not influence the results of the study. Conclusion: Decreased levels of TSP-1 and BMP-4 may serve as potential indices of lung cancer, with additional importance of low TSP-1 level as a marker of advanced stage of the disease.

## 1. Introduction

Lung cancer belongs to the most common carcinoma worldwide. It is the leading cause of cancer-related death in both genders [1,2]. Despite modern diagnostic techniques and new treatment options, the long-term survival only improved a little and prognosis still remains poor [3]. There are some known lung cancer biomarkers such as epidermal growth factor receptor (EGFR) mutations, anaplastic lymphoma kinase, *ROS* gene rearrangements or immunohistochemical expression of programmed death-ligand 1 (PD-L1) that significantly influenced clinical practice [4,5,6]. Better understanding of carcinogenesis and identification of some other biomarkers could have diagnostic and/or prognostic value. That is why a search of a new promising biomarkers plays important role in lung cancer diagnosis and monitoring. There are some laboratory and clinical data that may indicate that two other particles may join to still elongating list of biomarkers in the lung cancer patients. These are thrombospondin-1 (TSP-1) and bone morphogenetic protein-4 (BMP-4).

TSP-1 belongs to thrombospondins family, which consists of five homologous members TSP-1, TSP-2, TSP-3, TSP-4 and TSP-5 [7]. TSP-1, a filamentous glycoprotein ~70 nm long, is encoded by the *THBS1* gene in human [8] and has homotrimeric form [9]. TSP-1 is produced by many different cell types, including thrombin-stimulated platelets and endothelial cells [10]. This protein participates in the cell-to-cell and cell-to-matrix interactions and thus plays important role in tumor microenvironment for cancer development and metastasis formation [11]. Solid tumors consist not only of active proliferating carcinoma cells but also of other cell types that create the proper microenvironment promoting tumor progression [12]. TSP-1 in the tumor microenvironment influences the biology of multiple cell types associated with tumor growth and metastatic potential [13]. TSP-1 binds to the cell surface receptor CD47, known also as integrin-associated protein, through the globular carboxy-terminal cell-binding domain. TSP-1/CD47interaction plays important role in tumor progression through the influence on the tumor microenvironment [12,14]. However, the role of TSP-1 in tumor progression is still disputable and there are the studies suggesting opposite effects of TSP-1 depending on the molecular and cellular composition of the microenvironment [15].

TSP-1 is also known as anti-angiogenic factor. TSP-1 may inhibit angiogenesis through direct effects on endothelial cell migration, proliferation, survival and apoptosis [16]. It participates also in regulation of vascular endothelial growth factor (VEGF) bioavailability and activity [15]. TSP-1 directly inhibits VEGF by binding the growth factor and internalizing it via low-density lipoprotein receptor-related protein 1 (LPR1) [17]. TSP-1 is also involved in modulation of tumor immunity [15], including immunosuppressive effect on T cells [11].

BMP-4 is extracellular signaling molecule belonging to the transforming growth factor beta (TGFβ) superfamily. BMP-4 is involved in many important processes from bone and cartilage formation through participation in the regulation of cell proliferation and mobility to apoptosis and angiogenesis [18]. BMP-4 also participates in adipogenesis. It stimulates the differentiation of preadipocytes into mature adipocytes [19]. BMP-4 can inhibit cell growth and stimulate migration, invasion and epithelial-mesenchymal transition [18]. BMP-4 also contributes to cancer-induced immunosupression. In one study BMP-4 increased the expression of the T cell co-inhibitory receptor ligand PD-L1, leading to CD8+ T cell-mediated immunosupression [20]. BMP-4 induces expression of metalloproteinases in neoplastic cells and fibroblasts and in this way could stimulate the matrix degradation capacity of neoplastic cells or activate fibroblasts surrounding tumors to higher expression of matrix metalloproteinases [21]. Genetic variants in BMP-4 in the form of single nucleotide polymorphism (SNP) can be responsible for differences in BMP-4 production and activity.

In vitro study showed that BMP-4 suppressed in paracrine mechanism tumor angiogenesis process via the induction of TSP-1. Effectiveness of BMP-4 in inhibition of the tumor growth was lower in TSP-1 knockdown cancer cells. In this study BMP-4 also reduced VEGF expression in vivo in a TSP-1 dependent manner [22]. This observation confirmed the importance of BMP-4/TSP-1 loop in tumor angiogenesis. It has been found that BMP4- nuclear factor of activated T-cells, cytoplasmic 1-TSP-1 axis played important role in lung stem cell differentiation directed by endothelial cells [23].

There is little information regarding the significance of the combination of serum levels of these two proteins, i.e., TSP-1 and BMP-4, and BMP-4 polymorphism in the development and progression of lung cancer. Taking into account the combined influence of BMP-4 and TSP-1 on cancer cells [22,23] we aimed to investigate the serum levels of TSP-1 and BMP-4 together with SNP of gene coding for BMP-4 (rs1957860) in lung cancer patients in relation to histological type, lymph nodes involvement and the presence of remote metastasis. We did not find any previous studies, in which serum levels of TSP-1 and BMP-4, and BMP-4 polymorphism were evaluated together in lung cancer patients.

## 2. Material and Methods

### 2.1. Study Groups

A total of 111 consecutive patients (76 men) with newly diagnosed lung cancer were enrolled in this study. The main inclusion criterion was histopathologically confirmed lung cancer. The patients with any other neoplasm history were excluded from the study. The other exclusion criteria included corticosteroid, biological or immunosuppressive treatment. The mean age of the patients was 65.98 ± 8.58 years. Mean body mass index (BMI) was 24.6 ± 4.6 kg/m^2^. There were 102 patients with non-small cell lung cancer (NSCLC), including squamous cell cancer in 36 patients, adenocarcinoma in 22 patients and non-specified (NOS) cancer in 44 patients, and 9 patients with small cell lung cancer (SCLC). Advanced stage of disease was diagnosed in the majority of patients. There were 48 patients in stage IV, 33 patients with stage IIIB, 18 with stage IIIA, 6 with stage II (three-IIA and three-IIB) and 6 with stage I (four-IB, two-IA). The majority of patients were smokers with mean pack-years 30.83 ± 13.84 and only four patients were never smokers.

The control group consisted of 61 healthy subjects recruited from Regional Centre of Transfusion Medicine and Blood Bank.

### 2.2. ELISA Testing

Blood samples were collected from fasting subjects in the morning. After centrifugation for 10 min at 1467 RCF the serum was extracted and stored at −80 °C. TSP-1 and BMP-4 serum levels were measured using the enzyme-linked immunosorbent assay (ELISA) method. The following kits were used: Human BMP-4 (Thermo Fisher Scientific, Waltham, MA, USA) and Human Thrombospondin-1 (R&D Systems, Minneapolis, MN, USA). The tests were performed according to the manufacturer’s specifications. The ELISA microplate reader from MRXe Dynex Technologies (Chantilly, VA, USA) was used.

### 2.3. DNA Extraction and Genes Polymorphism Study

Genomic DNA was extracted from EDTA-anticoagulated peripheral blood using innuPREP Blood DNA Master Kit (Analytik Jena, Jena, Germany) according to the manufacturer’s recommendations and stored at −20 °C until required.

Genotyping was performed by the LightSNiP typing assay (TIB-MolBiol, Berlin, Germany) on a LightCycler 480 Real-Time PCR system (Roche Diagnostics, Rotkreuz, Switzerland) following manufacturer’s instructions.

### 2.4. Statistical Analysis

Statistical analysis was performed using the CSS Statistica software for Windows (version 5.0). Spearman’s *r* correlation coefficient was used to assess the relationship between two variables and the Mann-Whitney U test was used to compare values between the two groups. Differences between samples were considered significant at *p* < 0.05.

## 3. Results

TSP-1 serum levels were significantly lower in the patients with lung cancer than in the controls (TSP-1: 10,109.2 ± 9581 ng/mL vs. 11,415.09 ± 9781 ng/mL *p* = 0.012), Figure 1. In addition, TSP-1 serum levels were lower in patients with advanced lung cancer stages III-IV than in patients with early stages I-II (9282.07 ± 4900.78 ng/mL vs. 16,933.60 ± 6299.02 ng/mL; *p* = 0.03), (Table 1, Figure 2). The comparison of patients without lymph nodes metastases (N0) and the patients with metastases in any group of lymph nodes (N1) revealed that TSP-1 serum levels were decreased in lung cancer patients with lymph nodes involvement (10,000.13 ± 9021.41 ng/mL vs. 18,497.75 ± 12,548.25 ng/mL; *p* = 0.01), (Figure 3). There were no differences in age, BMI or histopathological type between these groups (Table 2).

BMP-4 serum levels were significantly lower in the patients with lung cancer than in the control group (138.35 ± 62.59 pg/mL vs. 226.68 ± 135.86 pg/mL, *p* < 0.001), (Figure 4). In the case of BMP-4 there were no differences in BMP-4 levels related to lung cancer stages (Table 1) or lymph nodes involvement (Table 2).

There were no differences in BMP-4 or TSP-1 levels in NSCLC as compared with SCLC patients. There were no differences in BMP-4 or TSP-1 serum levels in different histological subtypes of NSCLC, especially between the two most common histopathological types of NSCLC: adenocarcinoma and squamous cell lung cancer patients. Differences neither in TSP-1 levels (12,414.18 ± 11,650.28 ng/mL vs. 10,683.84 ± 10,534.9 ng/mL *p* = 0.712) nor in BMP-4 (103.60 ± 60.55 pg/mL vs. 166,70 ± 62.59 pg/mL *p* = 0.655) were found (Table 3).

There was no correlation between TSP-1 and BMP-4 serum levels. There were no differences in BMP-4 and TSP-1 serum levels related to BMP-4 gene polymorphisms (Table 4).

## 4. Discussion

The main importance of our study is finding of decreased serum levels of TSP-1 and BMP-4 in lung cancer patients. Moreover, we showed that TSP-1 serum levels were lower in patients with advanced lung cancer stages and lymph nodes involvement. In the previous studies BMP-4 and TSP-1 in lung cancer patients were mostly studied separately [24,25,26,27].

Our results showing decreased serum levels of TSP-1 in lung cancer patients, are in agreement with some previous studies [24,25,26]. The importance of TSP-1 in lung cancer emphasized the studies in which many biomarkers were evaluated at the same time. In one study 70 cancer-related proteins were measured and the authors identified a panel of 10 serum biomarkers, including TSP-1, which distinguished lung cancer patients from the controls [28]. Newest study demonstrated that TSP-1 together with two other biomarkers: brain angiogenesis inhibitor 1 and E-Cadherin had higher sensitivity and specificity than classical biomarkers in early diagnosis of lung cancer [29].

We did not find any previous study in which lower TSP-1 serum correlated with lymph nodes involvement. Our findings could indicate the prognostic value of TSP-1. In primary resected NSCLC (*n* = 171) TSP-1 levels inversely correlated with survival [25]. In this study for each 10 units of increment of TSP-1 the risk of death decreased by 85% and the multivariate analysis confirmed that decreased TSP-1 was independent prognostic factor of overall survival. However, there was no correlation between TSP-1 serum levels and distant metastasis free survival. In another study a set of angiogenesis, inflammation and coagulation markers was analyzed in patients treated for advanced NSCLC. Only lower TSP-1 serum levels and higher VEGF were unfavorable prognostic factors in the examined group [26]. Moreover, previous immunohistochemical studies revealed that reduced expression of TSP-1 in resected tumor tissue was associated with shorter 5-year survival [30]. Low TSP-1 mRNA expression was also connected with poorer prognosis in advanced NSCLC [31]. Even though BMP-4 seems to be promising molecule in lung cancer diagnosis and monitoring, the majority of studies based on tissue expression or genetic examinations. We were not able to find any previous publication in which BMP-4 serum levels were examined in lung cancer patients.

Higher expression of BMP-4 mRNAs (2.3 times) in lung cancer than in normal lung tissue has been reported [32]. Newest in vitro studies confirmed the role of BMP-4 in lung cancer progression. In the BMP-4 knockdown lung cancer cell lines inhibition of cell growth, migration, invasion and metastases formation was observed [33].

Genetic polymorphisms of BMP-4 and BMP-4 tissue expression may serve as potential predictor of the chemotherapy response and prognosis. In the group of 938 patients with advanced NSCLC (stages III and IV) the 6007CC genotype variant of BMP-4 was related to higher chance of platinum-based chemotherapy response, but in the patients with high BMP-4 protein expression chemotherapy resistance was observed more commonly [27].

The newest experimental studies revealed overexpression of BMP-4 in the NSCLC cells with acquired EGFR-tyrosine kinase inhibitor resistance. In addition, the authors indicated that BMP-4 mechanism of action was associated with activation of neoplastic cells energy metabolism [34]. Moreover, in vitro studies indicated that BMP-4 could be potential therapeutic target in the treatment of lung cancer patients [35].

Experience in the other neoplastic diseases indicate the BMP-4 importance. In breast cancer BMP-4 promotes the migration and invasion of cancer cells, at least in part by up-regulating the expressions of matrix metalloproteinase-1 and chemokine receptor type 4 [36].

Studies in non-neoplastic diseases indicate on the relationship between BMP-4 and obesity with metabolic syndrome. Higher BMP-4 serum levels in obese patients, positively correlating with BMI, waist circumference, waist-to-hip ratio and fasting insulin have been found [37]. Weight lost associated with gastric bypass led to significant decrease of BMP-4 [38].

It is not easy to explain, why the decreased levels of these two markers: TSP-1 and BMP-4 might promote tumor development, because the role in cancer progression of these two analyzed markers remains unclear. A number of studies confirm anti-tumor functions of TSP-1 and BMP-4 [13,17,39,40,41].

In the case of TSP-1, as mentioned in introduction, majority of available literature indicate the role of tumor microenvironment [13], especially TSP-1/CD47 interaction seems to play key role [12,14]. In addition, in premalignant lung tumors TSP-1 participates in suppression of Kras-induced tumorigenesis [42]. TSP-1 is also identified as neovascularization blocker, which could slow down malignant tumor growth [17]. That is why some authors consider TSP-1 to be the anti-metastatic protein [39].

BMP-4 is involved in inhibition of metalloproteinase-9, which is locally produced by tumor and through the degradation of the extracellular matrix, increased proliferation and invasiveness of tumor [43]. In addition, BMP-4 decreased the tumorigenic potential of human brain-initiating cells [40]. In animal model of breast cancer, it has been shown that BMP4 blocked metastasis formation through the activation of canonical BMP4-SMAD7 signaling, what might suggest that targeting the BMP4-SMAD7 signaling axis presents a novel therapeutic strategy [41].

Numerous data indicate that SNP variants could be implicated in the development of various cancer types [44]. Genetic variants of BMP-4 could lead to the changes in the local production of BMP-4 or in its effectiveness [27]. In lung cancer patients BMP-4 genetic variants influenced platinum-based chemotherapy response and prognosis [45].

Table 5 shows summarization of the previous laboratory and clinical studies indicating the role of TSP-1 and BMP-4 in the lung cancer development. Our study adds to the existing knowledge the findings that not only both TSP-1 and BMP-4 levels as measured by ELISA in the serum of the patients are decreased in lung cancer, but also that TSP-1 is associated with more advanced lung cancer stages and lymph nodes involvement. Even though our results appeared negative the correlation between TSP-1 and BMP-4 serum levels, as well as the possible association between BMP-4 gene polymorphism and serum levels of this biomarker have not been studied before.

## 5. Conclusions

The study confirms a role of decreased serum expression of TSP-1 and BMP-4 as the markers of lung cancer, with the lowest TSP-1 concentrations associated with more advanced lung cancer stages and with lymph nodes involvement.

## Figures and Tables

**Figure 1 jcm-10-03859-f001:**
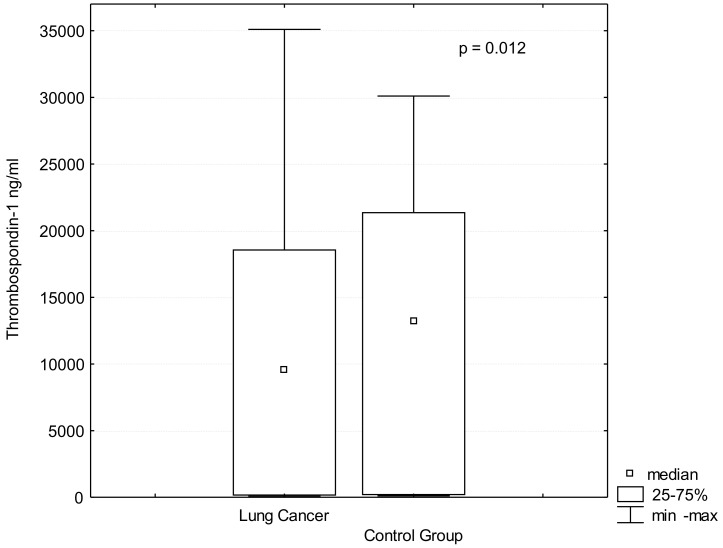
TSP-1 (Thrombospondin-1) serum levels in lung cancer and control group.

**Figure 2 jcm-10-03859-f002:**
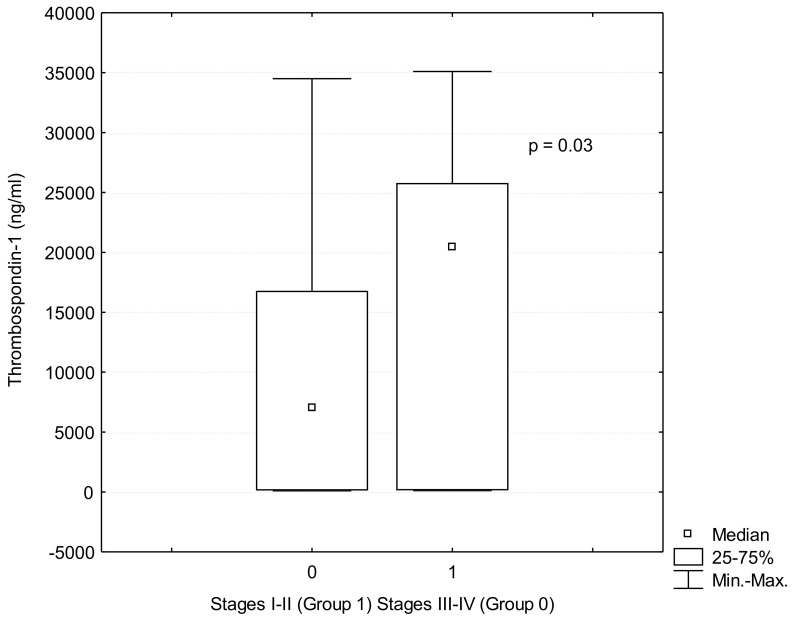
TSP-1 (Thrombospondin-1) serum levels in lung cancer patients in stages I-II (Group 1) and in stages III-IV (Group 0).

**Figure 3 jcm-10-03859-f003:**
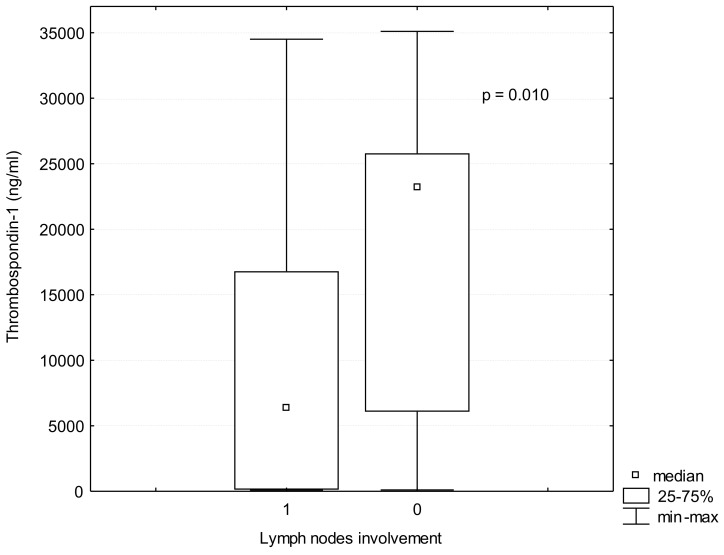
TSP-1 (Thrombospondin-1) serum levels in lung cancer patients with lymph nodes involvement (Group 1) and without lymph nodes involvement (Group 0).

**Figure 4 jcm-10-03859-f004:**
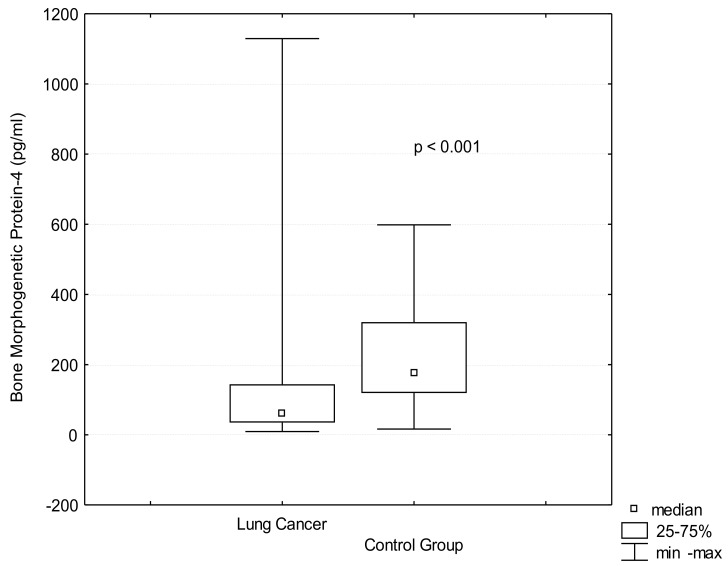
BMP-4 serum levels in lung cancer and control group. BMP-4 (bone morphogenetic protein-4).

**Table 1 jcm-10-03859-t001:** The comparison of lung cancer patients in different stages of the disease.

Parameters	Stage I (*n* = 6)	Stage II (*n* = 6)	Stage III (*n* = 51)	Stage IV (*n* = 48)	*p*
Age (years)	71.33 ± 7.71	70.33 ± 10.32	66.64 ± 8.71	64.33 ± 8.49	NS
BMI (kg/m^2^)	24.87 ± 4.53	23.66 ± 2.21	24.75 ± 5.00	24.48 ± 4.41	NS
TSP-1 (ng/mL)	19,355.5 ± 7914.07	14,510.50 ± 5000.2	9850.82 ± 4500.2	9035.18 ± 5124.9	NS
BMP-4 (pg/mL)	132.92 ± 32.92	128.60 ± 43.06	133.87 ± 65.38	144.73 ± 60.55	NS
Stages I and II (*n* = 12)			Stages III and IV (*n* = 99)		
Age (years)	70.83 ± 8.70		65.39 ± 8.42		NS
BMI (kg/m^2^)	24.27 ± 3.45		24.62 ± 4.70		NS
TSP-1 (ng/mL)	16,933.60 ± 6299.02		9282.07 ± 4900.78		*p* = 0.03
BMP-4 (pg/mL)	140.96 ± 55.54		143.29 ± 63.99		NS

TSP-1 (Thrombospondin-1), BMP-4 (bone morphogenetic protein-4), BMI (body mass index).

**Table 2 jcm-10-03859-t002:** The comparison of lung cancer patients without metastases in lymph nodes with patients with any lymph nodes involvement.

Parameters	Patients without Metastases in Lymph Nodes (*n* = 12)	Patients with Metastases in Any Lymph Nodes Group (*n* = 99)	*p*
Age (years)	67.66 ± 7.59	65.53 ± 8.62	NS
BMI (body mass index) (kg/m^2^)	24.99 ± 4.43	24.53 ± 4.61	NS
TSP-1 (ng/mL)	18,497.75 ± 12,548.25	10,000.13 ± 9021.41	*p* = 0.01
BMP-4 (pg/mL)	98.56 ± 62.59	143.82 ± 62.21	NS

TSP-1 (Thrombospondin-1), BMP-4 (bone morphogenetic protein-4), BMI (body mass index).

**Table 3 jcm-10-03859-t003:** TSP-1 and BMP-4 serum levels in different histopathological types of lung cancer.

Parameters	NSCLC (*n* = 102)	SCLC (*n* = 9)	*p*
TSP-1 (ng/mL)	10,093.04 ± 10,716.06	10,292.44 ± 10,491.16	NS
BMP-4 (pg/mL)	140.67 ± 60.55	114.23 ± 82.96	NS
Squamous Cell Lung Cancer (*n* = 36)		Adenocarcinoma (*n* = 22)	
TSP-1 (ng/mL)	10,683.84 ± 10,534.9	12,414.18 ± 11,650.28	NS
BMP-4 (pg/mL)	166.70 ± 62.59	103.60 ± 60.55	NS

TSP-1 (Thrombospondin-1), BMP-4 (bone morphogenetic protein-4), NSCLC (non small cell lung cancer), SCLC (small cell lung cancer).

**Table 4 jcm-10-03859-t004:** TSP-1 and BMP-4 serum levels and BMP-4 polymorphism.

Parameters	rs1957860			*p*
	CT (*n* = 52)	TT (*n* = 36)	CC (*n* = 25)	
BMP-4 serum levels (pg/mL)	137.82 ± 62.21	154.14 ± 76.12	140.81 ± 57.25	NS
TSP-1 serum levels (ng/mL)	11,982.26 ± 10,091.37	10,672.13 ± 9298.92	13,412.76 ± 12,421.90	NS

TSP-1 (Thrombospondin-1), BMP-4 (bone morphogenetic protein-4), CT (cytosine, thymine), TT (thymine, thymine), CC (cytosine, cytosine).

**Table 5 jcm-10-03859-t005:** TSP-1 and BMP-4 studies in lung cancer patients.

Biomarker	Material/Method	Results/Conclusion	Author
TSP-1	Plasma of lung cancer patient/ELISA	Down-regulation of expression of TSP-1/potential usefulness in lung cancer diagnosis	Zhang L. et al., 2019 [29]
TSP-1	Lung metastatic tumors/gene expression signature	Up-regulation in lung metastes of osteosarcoma/contribution of TSP-1 to lung metastases	Hu C. et al., 2017 [46]
TSP-1	Lung cancer patients undergoing surgical treatment/ELISA	TSP-1 is inversely correlated with survival rates/usefulness of TSP-1 as prognostic marker	Rouanne M. et al., 2016 [25]
TSP-1	Microarray databases/systematic-analysis	Underexpression in lung cancer/TSP-1 as tumor suppressor in lung adenocarcinoma	Weng T.Y. et al., 2016 [47]
TSP-1	Lung cancer specimens/immunochemistry	Lower expression of TSP-1 in late than in early stage of lung cancer/high expression of TSP-1 may inhibit tumor development	Chen Z.J. et al., 2009 [48]
TSP-1	Plasma of advanced lung cancer patients/ELISA	Decreased TSP-1 levels than in healthy controls/TSP-1 may increase tumor angiogenesis	Dudek A.Z. and Mahaseth H. 2005 [24]
TSP-1	Different tumors/review	TSP-1 suppresses tumor progression and inhibits angiogenesis	Lawler J. 2002 [49]
TSP-1	Resected lung cancer specimens/immunochemistry	Reduced expression of TSP-1 is associated with decreased 5-year survival/usefulness of TSP-1 as prognostic marker	Yamaguchi M. et al., 2002 [30]
BMP-4	Lung cancer specimens/qPCR and immunochemistry; survival of the patients	Increased BMP-4 levels in advanced stages of lung cancer and in the patients with shortened survival /usefulness of BMP-4 as prognostic factor	Ju F.J. et al., 2019 [50]
BMP-4	Lung cancer specimens before chemotherapy/western blot assay	Higher BMP-4 expression increases risk of resistance to chemotherapy/the usefulness of BMP-4 tissue expression as a prognostic factor	Xian S. et al., 2014 [27]

TSP-1 (Thrombospondin-1), BMP-4 (bone morphogenetic protein-4), ELISA (enzyme linked immunosorbent assay), PCR (polymerase chain reaction).

## Data Availability

We exclude this statement.

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
