# Peer review of "Decreased Thrombospondin-1 and Bone Morphogenetic Protein-4 Serum Levels as Potential Indices of Advanced Stage Lung Cancer"

_jcm, 2021, doi:10.3390/jcm10173859_

Round 1

Reviewer 1 Report

The authors have addressed my concerns and therefore I support the publication of this manuscript.

However, there are still some grammatical and typo errors that should be corrected. 

Author Response

Dear Editor,

Thank you very much for sending the manuscript Decreased thrombospondin-1 and bone morphogenetic protein-4 serum levels as potential indices of advanced stage lung cancer ( ID jcm-1322055) with Reviewer’s suggestions.

All changes are red highlighted. The answers to the Reviewers are as following.

Reviewer 1: Thank you very much for your supporting the publication. Grammatical and typo errors have been corrected.

I remain sincerely yours,

Monika Kosacka, MD, PhD

Department of Pulmonology and Lung Oncology, Wroclaw Medical University, Poland

Reviewer 2 Report

Kosacka et al investigated the role of TSP-1 and BMP-4 as biomarkers in lung cancer. Eventhough this is not a new topic, they have investigated the role of these molecules in patients which is still important. Please clarify the following concerns,

  1. Please clearly state the novel of the findings and methods employed?
  2. Please clearly state the exclusion and inclusion criteria of patients and provide more clinical data from these patients if it is available.
  3. Try to use publicly available data (like cBioportal) and generate a figure to see how your findings are connected to previous studies.

Author Response

Dear Editor,

Thank you very much for sending the manuscript Decreased thrombospondin-1 and bone morphogenetic protein-4 serum levels as potential indices of advanced stage lung cancer ( ID jcm-1322055) with Reviewer’s suggestions.

All changes are red highlighted. The answers to the Reviewers are as following.

Reviewer 2: Thank you for your valuable remarks. We added some  paragraphs according to your suggestions

  1. at the end of the discussion the following paragraph has been added: Our study adds to the existing knowledge the findings that not only both TSP-1 and BMP-4 levels as measured by ELISA in the serum of the patients are decreased in lung cancer, but also that TSP-1 is associated with more advanced lung cancer stages and  lymph nodes involvement.  And although our results appeared negative the correlation between TSP-1 and BMP-4 serum levels, as well as the possible association between BMP-4 gene polymorphism and serum levels of this biomarker have not been studied before.
  2. Inclusion and exclusion criteria were added as follows: A total of 111 consecutive patients ……The main inclusion criterion was histopathologically  confirmed  lung cancer. The patients with any other neoplasm history were excluded from the study. The other  exclusion criteria included  corticosteroid, biological or immunosuppressive treatment. 
  3. Data related to smoking status were added as follows: The majority of patients were smokers with mean pack-years 30.83±13.84 and only four patients were never smokers.
  4. In order to show how our results are connected with publicly available data: the Table 5 has been added (we have chosen the presentation in the Table rather than in a  Figure ).

I remain sincerely yours,

Monika Kosacka, MD, PhD

Department of Pulmonology and Lung Oncology, Wroclaw Medical University, Poland

This manuscript is a resubmission of an earlier submission. The following is a list of the peer review reports and author responses from that submission.

Round 1

Reviewer 1 Report

In the present manuscript the authors propose the decreased serum levels of TSP-1 and BMP-4 as prognostic markers of advanced stage of lung cancer. This is an interesting paper which highlights the prognostic value of two proteins the levels of which can be easily detected in patients’ serum through ELISA.

However, the authors should provide, based on the existing literature, a possible explanation why the decreased levels of these two markers might promote tumor development.

Minor revisions:

  • There are multiple mistakes regarding the labelling of the Figures/tables in the results section. For example, in the second paragraph Fig. 3 must be Fig. 4 and in the last paragraph Table 2 must be Table 4.
  • In addition, in pages 13-15 two figures are labelled as Fig 2.
  • In the correct Fig. 2 (page 14) the labelling in the x axis of the graph is wrong (0/1 does not correspond to the actual groups).
  • The language needs to be improved
